# A Random Walk Based Cluster Ensemble Approach for Data Integration and Cancer Subtyping

**DOI:** 10.3390/genes10010066

**Published:** 2019-01-18

**Authors:** Chao Yang, Yu-Tian Wang, Chun-Hou Zheng

**Affiliations:** 1College of Computer Science and Technology, Anhui University, Hefei 230601, Anhui, China; yiwaiyc@gmail.com; 2School of Software Engineering, Qufu Normal University, Qufu 273165, Shandong, China; wytfuture@gmail.com; 3Co-Innovation Center for Information Supply & Assurance Technology, Anhui University, Hefei 230601, Anhui, China

**Keywords:** cluster ensemble, random walk, refined similarity, cancer subtypes

## Abstract

Availability of diverse types of high-throughput data increases the opportunities for researchers to develop computational methods to provide a more comprehensive view for the mechanism and therapy of cancer. One fundamental goal for oncology is to divide patients into subtypes with clinical and biological significance. Cluster ensemble fits this task exactly. It can improve the performance and robustness of clustering results by combining multiple basic clustering results. However, many existing cluster ensemble methods use a co-association matrix to summarize the co-occurrence statistics of the instance-cluster, where the relationship in the integration is only encapsulated at a rough level. Moreover, the relationship among clusters is completely ignored. Finding these missing associations could greatly expand the ability of cluster ensemble methods for cancer subtyping. In this paper, we propose the RWCE (Random Walk based Cluster Ensemble) to consider similarity among clusters. We first obtained a refined similarity between clusters by using random walk and a scaled exponential similarity kernel. Then, after being modeled as a bipartite graph, a more informative instance-cluster association matrix filled with the aforementioned cluster similarity was fed into a spectral clustering algorithm to get the final clustering result. We applied our method on six cancer types from The Cancer Genome Atlas (TCGA) and breast cancer from the Molecular Taxonomy of Breast Cancer International Consortium (METABRIC). Experimental results show that our method is competitive against existing methods. Further case study demonstrates that our method has the potential to find subtypes with clinical and biological significance.

## 1. Introduction

With the efforts of the large-scale projects such as The Cancer Genome Atlas (TCGA) and the International Cancer Genome Consortium (ICGC) [1,2,3], a wealth of genome-scale molecular data are available and easy to access. The multiple types of omics data from genomes, transcriptomes, proteome, and epigenomes enable researchers to embrace great opportunities and possibilities to explore a more comprehensive view into cancer informatics, such as drug target prediction [4,5], diver gene identification [6,7,8], and so on.

One essential topic in oncology is cancer subtyping, whereby tumors are divided into clinically and biologically relevant subtypes, which could offer insight into tumor progression and provide personalized treatment. However, applying traditional clustering algorithms on a single data type—like gene expression data—does not obtain satisfactory results such as deriving subtypes associated with clinical phenotype [9]. These unsatisfactory results indicate the limitations of expression-based analysis for cancer subtyping. Since different types of molecular data contain information in various aspects that may complement each other, it is beneficial for leveraging different types of omics data simultaneously [10]. Several integrative frameworks have been proposed and gained success [9,10,11,12,13,14].

A promising method for cancer subtyping is cluster ensemble [11,15,16]. It can merge individual clusterings (clustering results obtained from running diverse clustering algorithms, running different types of omics data, etc.) to a consensus to form one robust unit. More importantly, cluster ensemble can naturally be applied on multiple data types as an integrative method. However, traditional cluster ensemble mostly merges different clusterings using a co-association matrix, which measures the frequency of two instances clustering together [17]. In this coarse way, some important information—such as relations among clusters—may be lost after merging the base clusterings. Link-based cluster ensemble (LCE) [15] tries to solve this problem by considering the relationship among clusters in terms of the triplet. The triplet is a subgraph containing three vertices and two non-zero edges. The similarity between two clusters is measured based on the count of all triples between them. However, this only captures local structure since the triplet measures the similarity in a local range.

In this paper, we proposed a new method named Random Walk based Cluster Ensemble (RWCE) to deal with these problems (Figure 1). We first obtained a refined cluster-cluster similarity by using random walk on a network of clusters constructed with Jaccard similarity and applied a scaled exponential similarity kernel, which provided a more global view from the whole cluster network. We then generated a more informative instance-cluster association matrix by filling in the refined cluster-cluster similarity. A bipartite graph was modeled on this resulting matrix in which spectral clustering [18] was used to obtain the final partition. Experiments on six cancer type datasets from the TCGA and the Molecular Taxonomy of Breast Cancer International (METABRIC) breast cancer data set [19] showed that our RWCE was competitive compared with other methods. Further case study demonstrated that our method also had the power to find clinically and biologically relevant subtypes. The source code of RWCE can be found in Appendix A.

## 2. Materials and Methods

### 2.1. Datasets

In order to show the effectiveness of our method, we used six TCGA cancer types: kidney renal clear cell carcinoma (KIRC), glioblastoma multiforme (GBM), lung squamous cell carcinoma (LUSC), breast invasive carcinoma (BRCA), acute myeloid leukemia (LAML), and colon adenocarcinoma (COAD). The data were processed by PINS (perturbation clustering for data integration and disease subtyping) [20], which is an integrative clustering framework for cancer subtyping. Each cancer type had three molecular omics data types, namely mRNA expression, miRNA expression, and DNA methylation. In addition, the METABRIC breast cancer data set [19] was used for survival analysis. The METABRIC data set included a discovery cohort (997 patients) and a validation cohort (995 patients). Each of them had two molecular data types: mRNA expression and copy number variation data, which were downloaded from the European Genome-Phenome Archive [21] (https://ega-archive.org/).

### 2.2. Competitive Methods

To show the effectiveness of our method, we compared it with a traditional cluster ensemble method called consensus clustering (CC) [17] as the baseline, and three state-of-the-art methods called link-based cluster ensemble (LCE) [15], perturbation clustering for data integration and disease subtyping (PINS) [20], and entropy-based consensus clustering (ECC) [11].

### 2.3. Evaluation Metrics

We used cox log-rank *p*-value [22] for measuring the significance of the difference of survival distributions between subtypes. Normally, a *p*-value < 0.05 indicates statistical significance, and a lower *p*-value indicates a more significant difference. We also used the silhouette value to measure consistency within subtypes. The mean value of the silhouette was used as a measure of how tightly grouped all the data in the cluster were. Higher values of the silhouette indicate a well-divided clustering structure.

For survival analysis, we also used concordance index (CI) [23], which measures the consistency between the estimated risk and the real survival time. Higher CI value indicates better performance for survival analysis.

### 2.4. Methodology Overview of RWCE

Here, we sum up RWCE for cancer subtyping. Suppose we have three data types to use for clustering. There are three steps in the RWCE pipeline. Step 1: For each data type, *M* basic clusterings are generated using *K*-means with a number of clusters randomly chosen from 2 to $\sqrt{n}$, where *n* is the number of instances (Figure 1A). Note that in this step, we can use any clustering method, and in this paper, we fixed it to *K*-means. Step 2: These *M* basic clusterings are combined into a consensus clustering by RWCE refinement (Figure 1B), which we introduce in detail later. Step 3: Each data type follows the same operation as in Step 1 and Step 2, and we then have three consensus clusterings—$\pi_{1}^{*},\pi_{2}^{*},\pi_{3}^{*}$. At last, we use RWCE refinement again to combine each data type’s consensus clustering to get the final clustering result $\pi^{*}$ (Figure 1C).

### 2.5. Cluster Ensemble

Let X denote an omics data set such as gene expression data with *n* instances (or conditions, experiments, patients, and so on) and let *m* denote genes (or biomarkers and so on). A cluster ensemble is a set of M basic clustering solutions generated by different clustering algorithms or a single clustering algorithm with different parameters, which is represented as $\Pi =\left\{\pi^{\left(1\right)},\pi^{\left(2\right)},\ldots \pi^{\left(M\right)}\right\}$. Each clustering $\pi^{\left(m\right)}$ partitions X into $K_{m}$ crisp clusters, represented as $\pi^{\left(m\right)}=\left\{C_{1}^{\left(m\right)},C_{2}^{\left(m\right)},\ldots C_{K_{m}}^{\left(m\right)}\right\}$, with $C_{k}^{\left(m\right)}\cap C_{k^{\prime }}^{\left(m\right)}=\emptyset ,\;\forall k\neq k^{\prime }$, and $\cup_{k=1}^{K_{m}}C_{k}^{\left(m\right)}=X$. The cluster ensemble method then takes these clusterings Π as input and combines these solutions to produce the consensus clustering $\pi^{*}$ as the output. There are diverse ways for combing [24,25]. One can derive an instance-cluster binary (IC) matrix with ‘1’, indicating that instance belongs to that cluster, otherwise it is indicated as ‘0’. Then, a clustering algorithm or graph segmentation algorithm could be used on this matrix to get the consensus clustering solution [17].

### 2.6. RWCE Refinement

#### 2.6.1. Generating a Refined Instance-Cluster Association (RIC) Matrix

A problem remains when leveraging an IC matrix or another similarity matrix since they only summarize information at a coarse level. For example, in an IC matrix, only one element is ‘1’ for one instance in each clustering, and others are ‘0’. This may lead to sparsity and does not favor the similarity-based clustering algorithm. Accordingly, in LCE [15], an improved variation of the original IC matrix, refined cluster-association matrix (RM) is generated by modifying the zero entries of the IC matrix with the cluster-cluster similarity discovered by the link-based similarity algorithm. The results show that the refinement is helpful and works better than using the original matrix. However, the algorithm used for measuring the similarity among clusters through focusing on triple is limited in local view. In response, we proposed RWCE, which has a more global view in discovering cluster-cluster similarity. We put forward a refined instance-cluster association (RIC) matrix as a more informative variation of the original IC matrix. It is designed to replace the value of those hidden associations (‘0’) of the IC matrix with the refined cluster similarity. For each clustering $\pi^{\left(m\right)},\;m=1\ldots M$ and their corresponding clusters $C_{1}^{\left(m\right)},C_{2}^{\left(m\right)},\ldots C_{K_{m}}^{\left(m\right)}$ (where $K_{m}$ is the number of clusters in clustering $\pi^{\left(m\right)}$), the association $RIC\left(x_{i},C\right)\geq 0\;and\leq 1$ between instance $x_{i}\in X$ and cluster $C\in \{C_{1}^{\left(m\right)},C_{2}^{\left(m\right)},\ldots C_{K_{m}}^{\left(m\right)}\}$ is measured as follows:
(1)$$RIC\left(x_{i},C\right)=\{\begin{array}{c} 1\;if\;C=C_{*}^{\left(m\right)}\left(x_{i}\right)\\ \frac{sim\left(C,C_{*}^{\left(m\right)}\left(x_{i}\right)\right)}{\sum_{\forall C\in \pi^{\left(m\right)}ȬC\neq C_{*}^{\left(m\right)}\left(x_{i}\right)}sim\left(C,C_{*}^{\left(m\right)}\left(x_{i}\right)\right)}\times dc\;otherwise \end{array}$$
where $C_{*}^{\left(m\right)}\left(x_{i}\right)$ is a cluster label to which the instance $x_{i}$ belongs in clustering $\pi^{\left(m\right)}$. Moreover, $sim\left(C_{x},C_{y}\right)\in \left[0,1\right]$ measures the similarity between any two clusters $C_{x},C_{y}$, which can be calculated using the random-walk based similarity algorithms listed in Section 2.6.2, and dc is a hyperparameter that we empirically set as 1 (performance is robust to *dc*, thus we fixed it to 1 for the sake of explanation). In this way, we fill in the zero entries of the IC matrix with the normalized similarity between the clusters by using the following random-walk based similarity algorithm.

#### 2.6.2. Random-Walk Based Similarity Algorithm

We first constructed an original cluster-cluster similarity network by using the Jaccard index as follows:(2)$$J_{xy}=\frac{\left|L_{x}\cap L_{y}\right|}{\left|L_{x}\cup L_{y}\right|}$$
where $J_{xy}$ is an edge of the above similarity network between cluster $C_{x}$ and $C_{y}$, and $L_{x}$ and $L_{y}$ denote the set of samples of clusters $C_{x}$ and $C_{y}$, respectively. On this initial network, we applied random walk with restart:
(3)$$F_{t+1}=\alpha F_{t}A+\left(1-\alpha \right)F_{0}$$
where A is the adjacency matrix of the above-mentioned similarity network and $F_{0}$ is the IC matrix. (1−α) is the restart probability that the random walker may choose to teleport to the initial node. The random walk process runs iteratively until $F_{t+1}$ converges ($|F_{t+1}-F_{t}|<1\;\times 10^{-6}$). In consequence, the resulted $F_{t+1}$ is a real-valued instance-by-cluster association matrix instead of a binary value, on which we can measure the refined similarity between clusters using the scaled exponential similarity kernel:(4)$$sim\left(C_{i},C_{j}\right)=exp\left(-\frac{\rho^{2}\left(z_{i},z_{j}\right)}{2\sigma^{2}}\right)$$
where $z_{i}$ and $z_{j}$ are *i*-th column and *j*-th column of $F_{t+1}$, representing clusters $C_{i}$ and $C_{j}$, respectively, $\rho^{2}\left(z_{i},z_{j}\right)$ denotes the squared Euclidean distance between cluster $C_{i}$ and $C_{j}$, and σ is a parameter we set to 1.

#### 2.6.3. Applying Spectral Clustering to RIC

As a result, we obtained a refined and informative instance-cluster (RIC) matrix; RIC(i,j)≥0 and ≤1 is a degree that instance i belongs to cluster j. We then modeled a bipartite graph G=(V,W) based on RIC, where $V=V^{C}\cup V^{I}$. $V^{C}$ is the set of vertices, where each vertex corresponds to a cluster from ensemble Π; $V^{I}$ is the set of vertices, where each vertex corresponds to an instance from data set X. W denotes a set of weighted edges that can be defined as follows:(5)$$W\left(i,j\right)=0\;if\;vertices\;v_{i},v_{j}\in V^{C}\;W\left(i,j\right)=0\;if\;vertices\;v_{i},v_{j}\in V^{I}\;W\left(i,j\right)=W\left(j,i\right)=RIC\left(i,j\right)\;if\;vertices\;v_{i}\in V^{C}\;and\;v_{j}\in V^{I}$$

Note that W can be written as $W=\left[\begin{array}{cc} 0 & RIC^{T}\\ RIC & 0 \end{array}\right]$ equivalently. Given such a graph, spectral graph partitioning (SPEC) [18] was then used to generate the final partition of X, denoted as $\pi^{*}$. SPEC with normalized cut is simply described as follows. Given graph G=(V,W), it first calculated the degree matrix D with degrees of each node on the diagonal. It then computed the Laplacian matrix L=D−W. Next, the normalized Laplacian matrix $D^{-\frac{1}{2}}LD^{-\frac{1}{2}}$, with its K smallest eigenvalues $\lambda_{1}\ldots \lambda_{k}$ and their corresponding eigenvectors $u_{1},u_{2}\ldots u_{K}$, were obtained. Then, a matrix $U=\left[u_{1},u_{2}\ldots u_{K}\right]$ was formalized after being row normalized. At last, SPEC generated the final clustering result using K-means on U. More details can be found in [18]. We selected the number of clusters $k=arg{\;}_{i>1}^{max}eigengap\left(i\right)$, where $eigengap\left(i\right)=\lambda_{i+1}-\lambda_{i}$. To sum up, we called the process of operating on ensemble Π and getting the final clustering $\pi^{*}$ as RWCE refinement.

### 2.7. Integrating Multiple Types of Omics Data for Subtyping

Suppose we had *T* types of omics data to integrate. For each type of omics data $X_{t},\;t=1\ldots T$, we obtained the corresponding clustering result $\pi_{t}^{*}$, *t* = 1…*T*. Then, we treated these clustering results as a new ensemble $\Pi^{*\;}=\left\{\pi_{1}^{*},\pi_{2}^{*},\ldots \pi_{T}^{*}\right\}$. Finally, we used RWCE refinement again to $\Pi^{*\;}$ to get the final clustering result $\pi^{*}$ across all *T* data types.

## 3. Results

### 3.1. Evaluation on TCGA Cancer Data Sets

For each cancer type, we counted the number of significant survival analyss results based on three single molecular data types and the integration of the three data types.

According to Figure 2, our method outperformed other methods on both single data type and the integration. By integrating the three molecular data types, our method attained significant subtypes (*p*-value < 0.05) for all six cancer types (Table 1). This indicates the potential of leveraging multiple data types simultaneously for identifying meaningful subtypes and the power of RWCE as an integrative method.

Table 1 shows the cox log-rank *p*-value of RWCE on three molecular data types and their integration across six cancer types from TCGA. It indicates that RWCE is a good integrative method for combining multiple omics data for cancer subtype discovery.

In terms of silhouette value (Figure 3), our method still outperformed other methods, indicating good clustering performance at the data level.

### 3.2. A Case Study: Glioblastoma Multiforme

Our method found three GBM subtypes. The survival curves of them are shown in Figure 4.

From Figure 4, subtype 1 had a bad prognosis while subtype 3 had a favorable prognosis. Moreover, Figure 5 shows that patients from subtype 1 had a favorable response to temozolomide (TMZ), a drug commonly used to treat GBM, and subtype 3 consisted of slightly younger patients.

### 3.3. Evaluation on METABRIC Data Set

We also tested the performance of survival analysis on the METABRIC breast cancer data set. As seen in Table 2, our method outperformed other clustering methods and was comparable with the PAM50 analysis (a standard breast cancer signature). This indicates the potential of our method for finding subtypes with differential survival profiles.

## 4. Discussion and Conclusions

In this paper, a new cluster ensemble method named RWCE was introduced for clustering and integrating multiple omics data to discover meaningful cancer subtypes. A novel RIC matrix is used in RWCE that considers relationships among clusters, which contributes to a superior clustering performance in terms of silhouette value and cox log-rank *p*-value.

Moreover, RWCE can also be utilized as an integrative method to make use of diverse types of omics data together for identifying subtypes with differential survival profiles. Further case study on the GBM subtypes that RWCE generated showed that RWCE could find subtypes with differential drug reactions and age distributions.

Taken together, RWCE provides a new way of thinking by combining basic clusterings in the cluster ensemble method and integrating multiple data types. We hope RWCE can generalize well to identify meaningful subtypes in more cancer types for the improvement of diagnostic and therapeutic intervention, and this is what we will investigate in further work.

## Figures and Tables

**Figure 1 genes-10-00066-f001:**
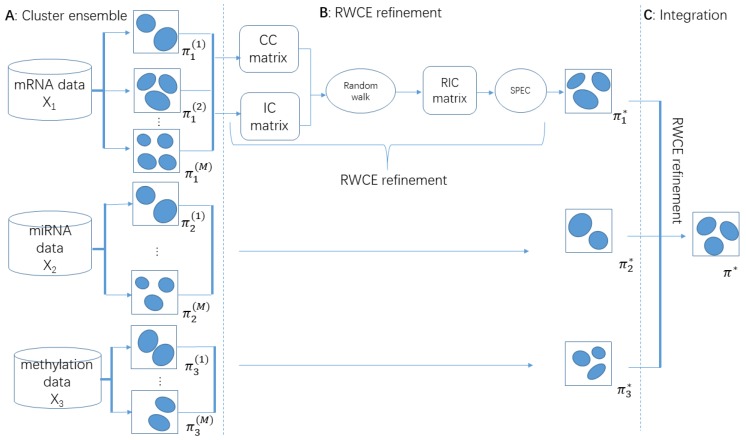
Schematic diagram of the Random Walk based Cluster Ensemble (RWCE) pipeline: (**A**) traditional clustering algorithm (here we used K-means) was applied to each molecular data type to obtain *M* basic clusterings. For each basic clustering, the cluster number was randomly chosen from 2 to √*n*; (**B**) each data type’s *M* clusterings were fused into one consensus clustering by RWCE refinement; (**C**) all data types’ consensus clusterings were fused into one final clustering using RWCE refinement again.

**Figure 2 genes-10-00066-f002:**
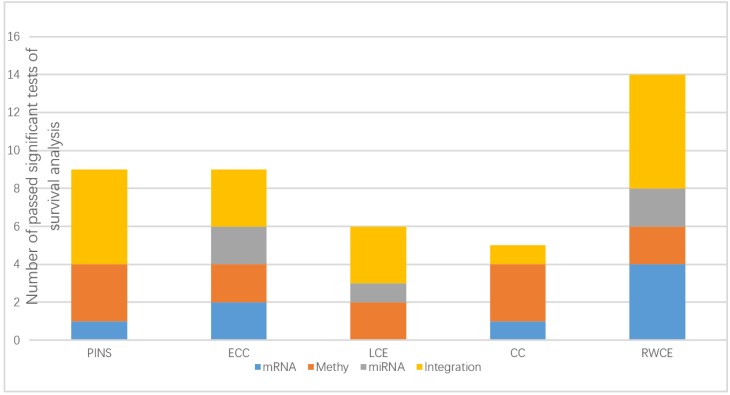
Stacked histogram displaying, for each clustering method (PINS: perturbation clustering for data integration and disease subtyping; ECC: entropy-based consensus clustering; LCE: link-based cluster ensemble; CC: consensus clustering; RWCE: random walk based cluster ensemble), the times it passed the significant tests (*p*-value < 0.05) of survival analysis on several molecular data types: mRNA expression data (mRNA), DNA methylation data (Methy), miRNA expression data (miRNA) and an integration of all three data types (integration).

**Figure 3 genes-10-00066-f003:**
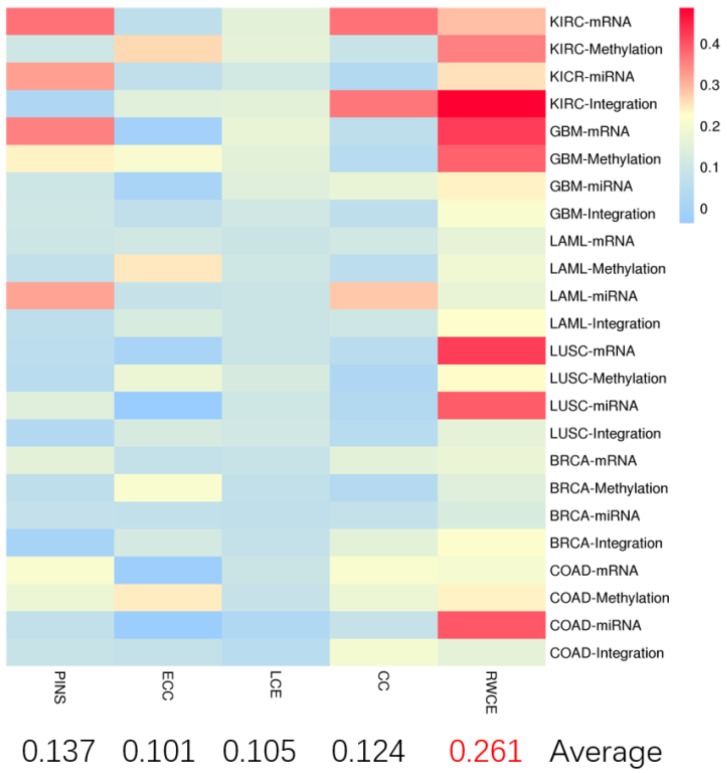
The heatmap for silhouette value on six TCGA datasets of different methods. KIRC-mRNA indicates mRNA expression data in KIRC was used. The same as the others.

**Figure 4 genes-10-00066-f004:**
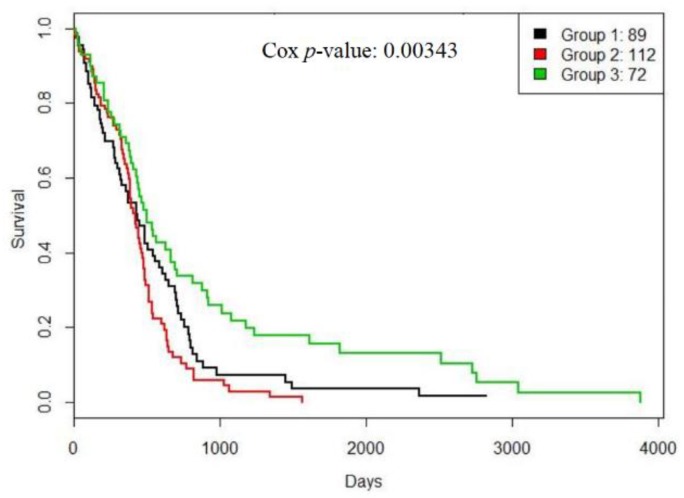
The survival curves for TCGA glioblastoma multiforme (GBM) subtypes generated by RWCE.

**Figure 5 genes-10-00066-f005:**
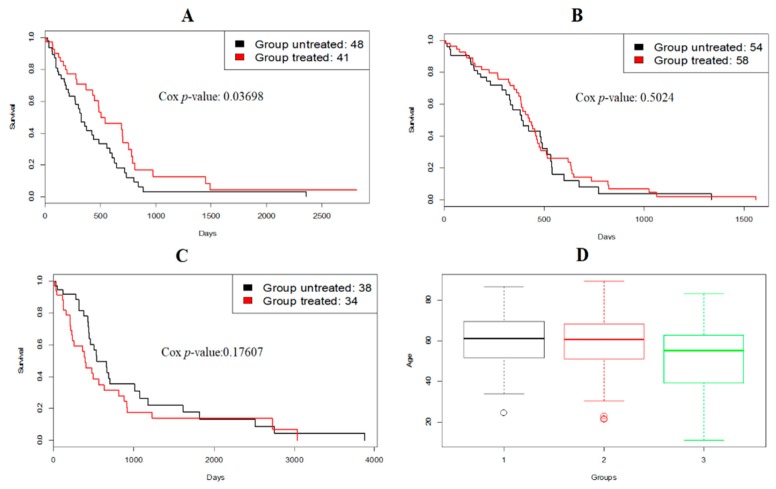
(**A**–**C**) Survival analysis of GBM patients for treatment with temozolomide (TMZ) in different subtypes generated by RWCE; (**D**) age distribution of GBM subtypes generated by RWCE.

**Table 1 genes-10-00066-t001:** Performance of RWCE on three molecular data types and their integration across six cancer types from The Cancer Genome Atlas (TCGA).

	mRNA	Methylation	miRNA	Integration
KIRC	**0.008(2)**	0.79397(3)	0.52883(2)	**0.00671(2)**
GBM	0.19041(2)	**0.00629(2)**	0.96568(2)	**0.00343 (3)**
LAML	**0.00272(8)**	0.58721(2)	**0.00119(8)**	**0.00158(2)**
LUSC	0.40747(3)	**0.04761(7)**	**0.01666(2)**	**0.00827(4)**
BRCA	**0.04193(2)**	0.58412(2)	0.15534(2)	**0.03006(2)**
COAD	**0.01058(2)**	0.68703(2)	0.81886(6)	**0.02818(3)**

KIRC (kidney renal clear cell carcinoma); GBM (glioblastoma multiforme); LAML (acute myeloid leukemia); LUSC (lung squamous cell carcinoma); BRCA (breast invasive carcinoma); COAD (colon adenocarcinoma). *p* < 0.05 is highlighted in bold.

**Table 2 genes-10-00066-t002:** Cox *p*-value and concordance index (CI) of subtypes discovered by PAM50, perturbation clustering for data integration and disease subtyping (PINS), consensus clustering (CC), entropy-based consensus clustering (ECC), link-based cluster ensemble (LCE), and our method on METABRIC data. For each discovery and validation cohort, we calculated the *p*-value and CI with respect to disease free survival (DFS) and overall survival of the patients. For each row, the best *p*-value (most significant) and the best CI (highest) are in red. The number of clusters in discovery and validation cohort are shown after the name of the clustering methods.

			PAM50 (5, 5)	PINS (14, 7)	CC (10, 8)	ECC (10, 10)	LCE (10, 8)	RWCE (6, 6)
Discovery	*p*-value	DFS	3.00 × 10^−11^	6.50 × 10^−10^	2.50 × 10^−5^	1.39 × 10^−1^	9.50 × 10^−1^	1.69 × 10^−9^
		Overall	8.50 × 10^−5^	1.90 × 10^−6^	8.10 × 10^−6^	5.59 × 10^−2^	4.42 × 10^−1^	4.16 × 10^−12^
	CI	DFS	0.620	0.634	0.598	0.521	0.506	0.594
		Overall	0.578	0.598	0.572	0.529	0.508	0.641
Validation	*p*-value	DFS	3.10 × 10^−9^	4.30 × 10^−5^	1.20 × 10^−2^	2.61 × 10^−1^	8.44 × 10^−2^	9.12 × 10^−5^
		Overall	2.90 × 10^−5^	033.80 × 10^−3^	7.90 × 10^−3^	1.66 × 10^−1^	3.53 × 10^−2^	9.13 × 10^−7^
	CI	DFS	0.636	0.589	0.572	0.521	0.520	0.560
		Overall	0.561	0.545	0.538	0.519	0.514	0.607

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
