# Peer review of "A Random Walk Based Cluster Ensemble Approach for Data Integration and Cancer Subtyping"

_genes, 2019, doi:10.3390/genes10010066_

Round 1
Reviewer 1 Report
The present paper suggests an alternative way of clustering based on certain molecular characteristics
However, it is rather unclear to me what is the potential clinical significance of this suggested method since generally clustering until now can mainly offer only scientific data and has not been widely applied in clinical practice
Author Response
Response to Reviewer 1.
Reviewer Comment: The present paper suggests an alternative way of clustering based on certain molecular characteristics
However, it is rather unclear to me what is the potential clinical significance of this suggested method since generally clustering until now can mainly offer only scientific data and has not been widely applied in clinical practice.
Response 1: It is true that clustering method for cancer subtyping is far from clinical practice. But some studies have shown good results. Perou [1] classified tumors on the basis of expression patterns by using hierarchical clustering method. Sørlie [2] further stratified the classifications described by Perou [1] and explored the clinical value of these breast cancer subtypes. Based on those results and further research, nowadays people classify breast tumor into five subtypes: Luminal-A, Luminal-B, HER2-enriched, Basal-like and Normal-like. For Luminal-A, as HER2 is negative, targeted drug therapy is not suitable. Endocrine treatment and surgical treatment are suitable for patient in this subtype. For Liminal-B, targeted drug therapy and endocrine treatment are suitable. Other subtypes have their suitable treatment options. The research is fruitful, the hospital use it as part of the reference for treatment. But the research requires priori knowledge in oncology. With more and more cancer omics data, some expert in CS try to do this job with machine learning method [3-5], The subtypes they got were similar to those used in the clinic, and some subtypes are novel or have positive response to drug treatment [4]. So I believe clustering method for cancer subtyping is a promising method for further clinical use although it is not perfect yet. Back to our paper, we did some further study to show that our method obtained subtypes with clinical differences. We obtained subtypes with statistically meaningful p value in survival analysis in all six cancer datasets we used. The GBM subtype 1 we found have positive response to TMZ drug treatment (P = 0.037). And subtypes show differences in age. Those differences between subtypes to some extent show clinical meaning, for example, patients in GBM subtype 1 could take TMZ for suitable treatment. As a researcher in CS, we are inadequate in the field of clinical analysis, but we hope our method could provide ideas for the development of integrative clustering framework. We will also do further case study for clinical use to seek for more clinical practice.
[1] Perou C M, Sørlie T, Eisen M B, et al. Molecular portraits of human breast tumours[J]. nature, 2000, 406(6797): 747.
[2] Sorlie T , Perou C M , Tibshirani R , et al. Gene expression patterns of breast carcinomas distinguish tumor subclasses with clinical implications[J]. Proceedings of the National Academy of Sciences, 2001, 98(19):10869-10874.
[3] Shen, R., A.B. Olshen, and M. Ladanyi, Integrative clustering of multiple genomic data types using a joint latent variable model with application to breast and lung cancer subtype analysis. Bioinformatics, 2009. 25(22): p. 2906-2912.
[4] Wang, B., et al., Similarity network fusion for aggregating data types on a genomic scale. Nature methods, 2014. 11(3): p. 333.
[5] Hofree, M., et al., Network-based stratification of tumor mutations. Nature methods, 2013. 10(11): p. 1108.

Reviewer 2 Report
In this manuscript the authors propose a novel ensemble method with improved performance. Overall, I think the methods are scientifically sound, but I had some serious concern with the presentation of the manuscript (eg. some methods were included in results, motivation of why and how the proposed method can help address shortfalls of competitors are lacking). I recommend the authors engage with a native english speaker to review the grammar and writing of the manuscript. I have attached detailed comments. I also would like to see code to reproduce the methods and the results of their experiment be made available in a github repo or on the author's website.

Author Response
Response to Reviewer 2 Comments
Thanks for pointing out grammar issues and typo mistakes. We have corrected them, you could see detail in cover letter.
Point 1: line 17: “However, many existing cluster ensemble methods neglect relations among clusters from an ensemble.” This sentence does not really explain the problem.
Response 1: We have explained the problem in detail: However, many existing cluster ensemble methods use co-association matrix to summarize the co-occurrence statistics of the instance-cluster, while the relationship in the integration is only encapsulated at a rough level. Moreover the relationship among clusters is completely ignored. Finding these missing associations could greatly expand the ability of cluster ensemble methods for cancer subtyping.
Point 2: line 40: I think the issue is not a “single data analysis” which is not really defined. It’s the analysis of expression-based analysis.
Response 2: We agree with your opinion. So we change “single data analysis” into “expression-based analysis”.
Point 3: line 50, 52: elaborate information in detail.
Response 3: In this coarse way, some important information such as relations among clusters may be lost after merging base clusterings.
The triplet is a subgraph containing three vertices and two non-zero edges. The similarity between two clusters are measured based on count of all triples between them. However, this could only capture local structure known as local view.
Point 4: line 75: “Each data type followed the same operation in RWCE before the final merging”. I would add this in the end, perhaps start by explaining RWCE on a single data.
Response 4: Thanks for your suggestion. We have deleted this sentence, and added it later : Step 3: Each data type followed same operation in step 1 and step 2, and we then have three consensus clustering .
Point 5: line 79: This detail here is confusing. You will delve into it later so don’t expand here.
Response 5: Thanks for your advice. So we delete this detail and explained it later.
Point 6: line 84: So step 1 and 2 are repeated for each data type and then step 2 is repeated again across data types. This is not what the text says. Based on the text, it sounds that you do 3 steps for each data type.
Response 6: Your understanding is right. We made a mistake in text. So we corrected it as follows: Step 1: For each data type, M basic clusterings were generated by using K-means with number of clusters randomly chosen from 2 to where n is the number of instances (Fig. 1A). Note that in this step, we can use any clustering method, and in this paper we fixed it to K-means. Step 2: These M basic clusterings were combined into a consensus clustering by RWCE refinement (Fig. 1B) which we will introduce in detail later. Step 3: Each data type followed the same operation in step 1 and step 2, and we then have three consensus clustering . At last we used RWCE refinement again to combine each data type’s consensus clustering to get the final clustering result (Fig. 1C).
Point 7: line 125: “?? is a hyperparameter that we empirically set as 1.” How is this set at 1 and what happens as this number goes up or down.
Response 7: Performance is robust to dc so we fixed it to 1 for the sake of explanation.
Point 8: Section 3.1 and 3.2 is in the wrong place.
Response 8: We have moved 3.1 and 3.2 into section 2.
Point 9: line 192: So 4*6 = 24 max total P value?
Response 9: Yes. The max counts is 24.
Point 10: In this manuscript the authors propose a novel ensemble method with improved performance. Overall, I think the methods are scientifically sound, but I had some serious concern with the presentation of the manuscript (eg. some methods were included in results, motivation of why and how the proposed method can help address shortfalls of competitors are lacking). I recommend the authors engage with a native English speaker to review the grammar and writing of the manuscript. I have attached detailed comments. I also would like to see code to reproduce the methods and the results of their experiment be made available in a GitHub repo or on the author's website.
Response 10:
1. Our proposed method is based on LCE. LCE measure cluster-cluster similarity based on triplet. The triplet is a subgraph containing three vertices and two non-zero edges. The similarity between two clusters are then measured based on count of all triples between them. This could only capture local structure known as local view. Instead of using triplet, we use random walk on whole cluster-cluster similarity network. Thus we could carry the full information about the similarity of each cluster to all others in the network.
2. We have offered source code in supplementary file.
